# SADGA: Structure-Aware Dual Graph Aggregation Network for Text-to-SQL

**Ruichu Cai**[1,2]**, Jinjie Yuan**[1]**, Boyan Xu**[1]*__**, Zhifeng Hao**[1,3]

[1] School of Computer Science, Guangdong University of Technology, Guangzhou, China
[2] Peng Cheng Laboratory, Shenzhen, China
[3] College of Science, Shantou University, Shantou, China
`cairuichu@gdut.edu.cn, yuanjinjie0320@gmail.com`
`hpakyim@gmail.com, zfhao@gdut.edu.cn`

## Abstract

The Text-to-SQL task, aiming to translate the natural language of the questions into SQL queries, has drawn much attention recently. One of the most challenging problems of Text-to-SQL is how to generalize the trained model to the unseen database schemas, also known as the cross-domain Text-to-SQL task. The key lies in the generalizability of (i) the encoding method to model the question and the database schema and (ii) the question-schema linking method to learn the mapping between words in the question and tables/columns in the database schema. Focusing on the above two key issues, we propose a *Structure-Aware Dual Graph Aggregation Network* (SADGA) for cross-domain Text-to-SQL. In SADGA, we adopt the graph structure to provide a unified encoding model for both the natural language question and database schema. Based on the proposed unified modeling, we further devise a structure-aware aggregation method to learn the mapping between the question-graph and schema-graph. The structure-aware aggregation method is featured with *Global Graph Linking*, *Local Graph Linking* and *Dual-Graph Aggregation Mechanism*. We not only study the performance of our proposal empirically but also achieved 3rd place on the challenging Text-to-SQL benchmark Spider at the time of writing.

## 1   Introduction

Structured Query Language (SQL) has become the standard database query language for a long time, but the difficulty of writing still hinders the non-professional user from using SQL. The Text-to-SQL task tries to alleviate the hinders by automatically generating the SQL query from the natural language question. With the development of deep learning technologies, Text-to-SQL has achieved great progress recently [5, 14, 33, 29].

Many existing Text-to-SQL approaches have been proposed for particular domains, which means that both training and inference phases are under the same database schema. However, it is hard for database developers to build the Text-to-SQL model for each specific database from scratch because of the high annotation cost. Therefore, cross-domain Text-to-SQL, aiming to generalize the trained model to the unseen database schema, is proposed as a more promising solution [11, 3, 4, 27, 7, 22, 19, 6]. The core issue of cross-domain Text-to-SQL lies in building the linking between the natural language question and database schema, well-known as the question-schema linking problem [11, 27, 19, 17, 35].

---

*Corresponding author

35th Conference on Neural Information Processing Systems (NeurIPS 2021).

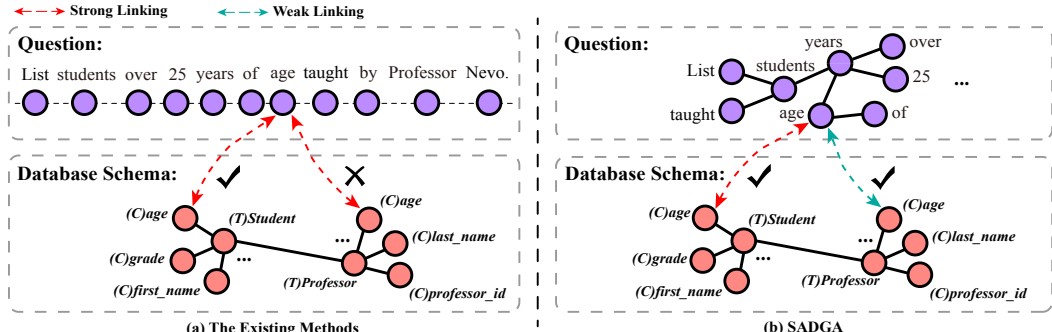

Figure 1: A toy example. The left part is about some existing approaches, e.g., IRNet [11], RATSQL [27], which usually treat the question as a sequence and apply the string matching method or attention mechanism to build the question-schema linking, causing that word "age" has a strong linking with both column "age" of table "Student" and column "age" of table "Professor" **(the red arrow)**. The right part is about our SADGA. We treat both the question and schema as the graph structure to eliminate the structure gap during linking, and use SADGA to explore the local structure to help build the linking, successfully removing the candidate linking between word "age" and column "age" of table "Professor" **(the green arrow)**.

There are two categories of efforts to solve the aforementioned question-schema linking problem — matching-based method [11] and learning-based method [27, 19, 17, 6]. IRNet [11] is a typical matching-based method, which uses a simple yet effective string matching approach to link question words and tables/columns in the schema. RATSQL [27] is a typical learning-based method, which applies a relation-aware transformer to globally learn the linking over the question and schema with predefined relations. However, both of the above two categories of methods still suffer from the problem of insufficient generalization ability. There are two main reasons for the above problem: first, the structure gap between the encoding process of the question and database schema: as shown in Figure 1, most of the existing approaches treat the question as a sequence and learn the representation of the question by sequential encoders [11, 3, 4] or transformers [27, 28, 19], while the database schema is the structured data whose representation is learned based on graph encoders [6, 3, 4] or transformers with predefined relations [27, 28]. Such the structure gap leads to difficulty in adapting the trained model to the unseen schema. Second, highly relying on predefined linking maybe result in unsuitable linking or the latent linking to be undetectable. Recalling the example in Figure 1, some existing works highly rely on the predefined relations or self-supervised learning on question-schema linking, causing the wrong strong linking between word "age" and column "age" of table "Professor" while based on the semantics of the question, word "age" should be only linked to the table "Student". Regarding the latent association, it refers to the fact that some tables/columns do not attend exactly in the question while they are strongly associated with the question, which is difficult to be identified. Such undetected latent association also leads to the low generalization ability of the model.

Aiming to alleviate these above limitations, we propose a Structure-Aware Dual Graph Aggregation Network (SADGA) for cross-domain Text-to-SQL to fully take advantage of the structural information of both the question and schema. We adopt a unified graph neural network encoder to model both the natural language question and schema. On the question-schema linking across question-graph and schema-graph, SADGA is featured with *Global Graph Linking*, *Local Graph Linking* and *Dual-Graph Aggregation Mechanism*. In the *Global Graph Linking* phase, the query nodes on question-graph or schema-graph calculate the attention with the key nodes of the other graph. In the *Local Graph Linking* phase, the query nodes will calculate the attention with neighbor nodes of each key node across the dual graph. In the *Dual-Graph Aggregation mechanism*, the above two-phase linking processes are aggregated in a gated-based mechanism to obtain a unified structured representation of nodes in question-graph and schema-graph. The contributions are summarized as follows:

- We propose a unified dual graph framework SADGA to interactively encode and aggregate structural information of the question and schema in cross-domain Text-to-SQL.

- In SADGA, the structure-aware dual graph aggregation is featured with *Global Graph Linking*, *Local Graph Linking* and *Dual-Graph Aggregation Mechanism*.

- We conduct extensive experiments to study the effectiveness of SADGA. Especially, SADGA outperforms the baseline methods and achieves 3rd place on the challenging Text-to-SQL benchmark Spider [2] [34] at the time of writing. Our implementation will be open-sourced at `https://github.com/DMIRLAB-Group/SADGA`.

## 2 Model Overview

We provide the overview of our proposed overall model in Figure 2. As shown in the figure, our model follows the typical encoder-decoder framework. There are two components of the encoder, Structure-Aware Dual Graph Aggregation Network (SADGA) and Relation-Aware Transformer (RAT) [27].

The proposed SADGA consists of dual-graph construction, dual-graph encoding and structure-aware aggregation. In the workflow of SADGA, we first construct the question-graph based on the contextual structure and dependency structure of the question, and build the schema-graph based on database-specific relations. Second, a graph neural network is employed to encode the question-graph and schema-graph separately. Third, the structure-aware aggregation method learns the alignment across the dual graph through two-stages linking, and the information is aggregated in a gated-based mechanism to obtain a unified representation of each node in the dual graph.

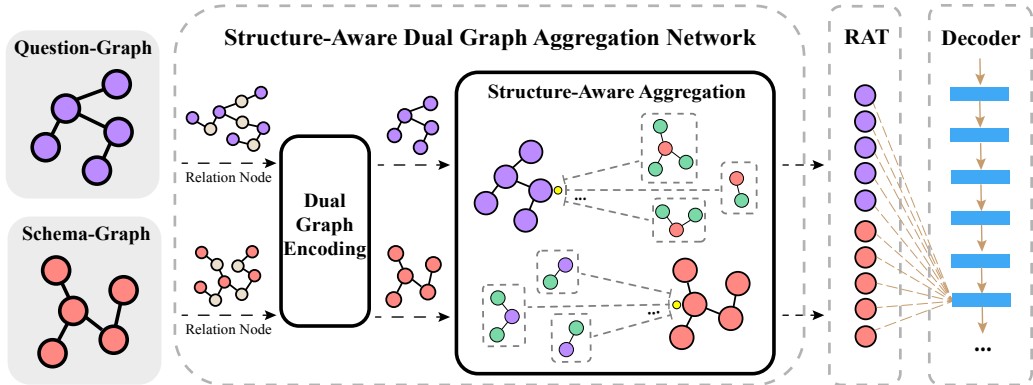

Figure 2: The overview of the proposed model.

RAT [27] tries to further unify the representations learned by our SADGA by encoding the question words and tables/columns with the help of predefined relations. RAT is an extension of Transformer [26], which introduces prior knowledge relations to the self-attention mechanism (see Appendix A.2). Different from the work of Wang et al. [27] with more than 50 predefined relations, the RAT of our work only uses 14 predefined relations, the same as those used by SADGA (Section 3.1). The small number of predefined relations also ensures the generalization ability of our method.

In the decoder, we follow the tree-structured architecture of Yin and Neubig [31], which transforms the SQL query to an abstract syntax tree in depth-first traversal order. First, we apply an LSTM [13] to output a sequence of actions that generates the abstract syntax tree; then, the abstract syntax tree is transformed to the sequential SQL query. These LSTM output actions are either schema-independent (the grammar rule) or schema-specific (table/column). Readers can refer to Appendix C for details.

## 3 Structure-Aware Dual Graph Aggregation Network

In this section, we will delve into the Structure-Aware Dual Graph Aggregation Network (SADGA), including Dual-Graph Construction, Dual-Graph Encoding and Structure-Aware Aggregation. The aggregation method consists of *Global Graph Linking*, *Local Graph Linking* and *Dual-Graph Aggregation Mechanism*. These three steps introduce the global and local structure information on question-schema linking. The details of each component are as follows.

---

[2] `https://yale-lily.github.io/spider`

## 3.1 Dual-Graph Construction

In SADGA, we adopt a unified dual-graph structure to model the question, schema and predefined linkings between question words and tables/columns in the schema. The details of the generation of question-graph, schema-graph and predefined cross-graph relations are as follows.

**Question-Graph** A question-graph can be represented by $\mathcal{G}_Q = (Q, R_Q)$, where the node set $Q$ represents the words in the question and the set $R_Q$ represents the dependencies among words. As shown on the left side of Figure 3, there are three different types of links, 1-order word distance dependency (i.e., two adjacent words have this relation), 2-order word distance dependency, and the parsing-based dependency. The parsing-based dependency is to capture the specific grammatical relationships among words in the natural language question, such as the clausal modifier of noun relationship in the left side of Figure 3, which is constructed by applying Stanford CoreNLP toolkit [20].

**Schema-Graph** Similarly, a schema-graph can be represented by $\mathcal{G}_S = (S, R_S)$, where the node set $S$ represents the tables/columns in the database schema and the edge set $R_S$ represents the structural relations among tables/columns in the schema. We use some typical database-specific relations, such as the primary-foreign key for column-column pairs. The right side of Figure 3 shows an example of a schema-graph, where we focus only on the linking from the column "professor_id".

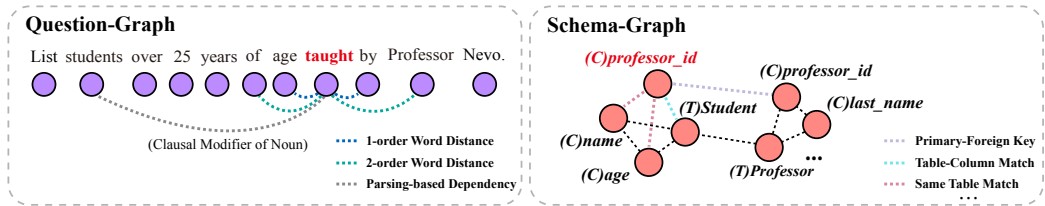

Figure 3: The construction of Question-Graph and Schema-Graph.

**Cross-Graph** We also introduce cross-graph relations to capture the connection between question-graph and schema-graph. There are two main rules to generate relations, exact string match and partial string match, which are borrowed from RATSQL [27]. We use these two rules for word-table and word-column pairs to build relations. Besides, for word-column pairs, we also use the value match relation, which means if the question presents a value word in the database column, there is the value match relation between the value word and the corresponding column. Note that these cross-graph relations are used only in Structure-Aware Aggregation (Section 3.3).

Moreover, all predefined relations in dual-graph construction are undirected. These relations are described in detail in Appendix B.

## 3.2 Dual-Graph Encoding

After the question-graph and schema-graph are constructed, we employ a Gated Graph Neural Network (GGNN) [18] to encode the node representation of the dual graph by performing message propagation among the self-structure before building the linking across the dual graph. The details of the GGNN we apply are presented in Appendix A.1. Inspired by Beck et al. [2], instead of representing multiple relations on edges, we represent the predefined relations of question-graph and schema-graph on nodes to reduce trainable parameters. Concretely, if node A and node B have a relation R, we introduce an extra node R into the graph and link R to both A and B using undirected edges. There is no linking across the dual graph in this phase. Each relation node is initialized to the learnable vector of the corresponding relation. In addition, we define three basic edge types for GGNN updating, i.e., bidirectional and self-loop.

## 3.3 Structure-Aware Aggregation

Following with dual-graph encoding, we devise a structure-aware aggregation method on question-schema linking between question-graph $\mathcal{G}_Q$ and schema-graph $\mathcal{G}_S$. The aggregation process is

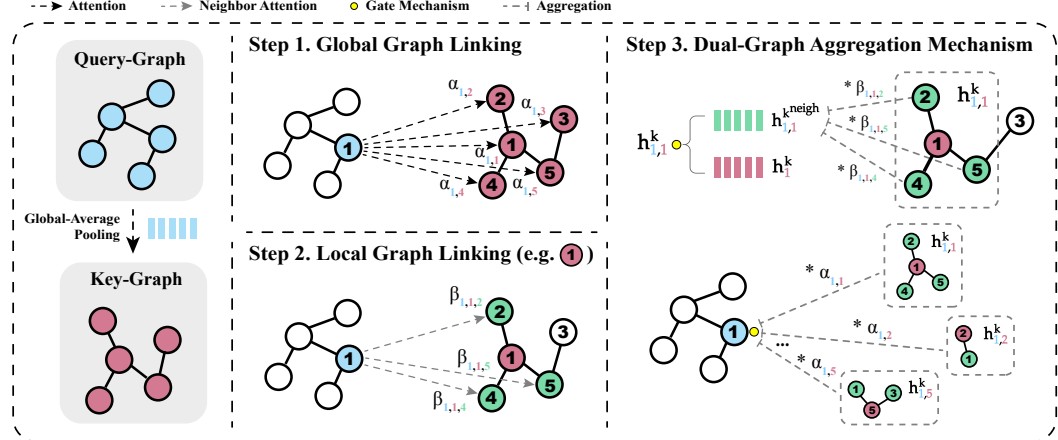

Figure 4: The Structure-Aware Aggregation procedure. We show the case when the 1st node in the query-graph acts as the query node. The query node attends to the key node and the neighbor nodes of the key node.

formulated as

$$\mathcal{G}_Q^{Aggr} = \text{GraphAggr}(\mathcal{G}_Q, \mathcal{G}_S), \quad \mathcal{G}_S^{Aggr} = \text{GraphAggr}(\mathcal{G}_S, \mathcal{G}_Q). \tag{1}$$

As shown in Eq. 1, the structure-aware aggregation method is applied to aggregate information from schema-graph $\mathcal{G}_S$ and question-graph $\mathcal{G}_Q$ to the other graph, respectively. We illustrate the detailed approach in the manner of query-graph $\mathcal{G}_q$ and key-graph $\mathcal{G}_k$, i.e.,

$$\mathcal{G}_q^{Aggr} = \text{GraphAggr}(\mathcal{G}_q, \mathcal{G}_k). \tag{2}$$

Let $\{\boldsymbol{h}_i^q\}_{i=1}^m$ be a set of node embedding in the query-graph $\mathcal{G}_q$ and $\{\boldsymbol{h}_j^k\}_{j=1}^n$ be a set of node embedding in the key-graph $\mathcal{G}_k$, which both learned by dual-graph encoding. Figure 4 shows the whole procedure of the structure-aware aggregation method regarding how the information from the key-graph is utilized to update the query-graph at the global and local structure level. First, we use global-average pooling on the node embedding $\boldsymbol{h}_i^q$ of query-graph $\mathcal{G}_q$ to get the global query-graph embedding $\boldsymbol{h}_{glob}^q$. Then, in order to capture globally relevant information, the key node embedding $\boldsymbol{h}_j^k$ is updated as follows:

$$\boldsymbol{h}_{glob}^q = \frac{1}{m} \sum_{i=1}^m \boldsymbol{h}_i^q, e_j = \theta\left(\boldsymbol{h}_{glob}^q{}^T \boldsymbol{W}_g \boldsymbol{h}_j^k\right), \tag{3}$$

$$\boldsymbol{h}_j^k = (1 - e_j)\boldsymbol{W}_{qg}\boldsymbol{h}_{glob}^q + e_j\boldsymbol{W}_{kg}\boldsymbol{h}_j^k, \tag{4}$$

where $\boldsymbol{W}_g$, $\boldsymbol{W}_{qg}$, $\boldsymbol{W}_{kg}$ are trainable parameters and $\theta$ is a sigmoid function. $e_j$ represents the relevance score between the $j$-th key node and the global query-graph. The above aggregation process is inspired by Zhang et al. [38]. Our proposed structure-aware aggregation method further introduces the global and local structural information through three primary phases, including *Global Graph Linking*, *Local Graph Linking* and *Dual-Graph Aggregation Mechanism*.

**Global Graph Linking**   *Global Graph Linking* is to learn the linking between each query node and the global structure of the key-graph. Inspired by the relation-aware attention [27], we calculate the global attention score $\alpha_{i,j}$ between query node embedding $\boldsymbol{h}_i^q$ and key node embedding $\boldsymbol{h}_j^k$ as follows:

$$s_{i,j} = \sigma\left(\boldsymbol{h}_i^q \boldsymbol{W}_q \left(\boldsymbol{h}_j^k + \boldsymbol{R}_{ij}^E\right)^T\right), \alpha_{i,j} = \text{softmax}_j\left\{s_{i,j}\right\}, \tag{5}$$

where $\sigma$ is a nonlinear activation function and $\boldsymbol{R}_{ij}^E$ is the learned feature to represent the predefined cross-graph relation between $i$-th query node and $j$-th key node. The cross-graph relations have already been introduced in Cross-Graph of Section 3.1.

**Local Graph Linking**    *Local Graph Linking* is designed to introduce local structure information on dual graph linking. In this phase, the query node calculates the attention with neighbor nodes of the key node across the dual graph. Specifically, we calculate the local attention score $\beta_{i,j,t}$ between $i$-th query node and $t$-th neighbor node of $j$-th key node, formulated as

$$o_{i,j,t} = \sigma \left( \boldsymbol{h}_i^q \boldsymbol{W}_{nq} \left( \boldsymbol{h}_t^k + \boldsymbol{R}_{it}^E \right)^T \right), \beta_{i,j,t} = \text{softmax}_t \{o_{i,j,t}\} \, (t \in \mathcal{N}_j), \quad (6)$$

where $\mathcal{N}_j$ represents the neighbors of the $j$-th key node.

**Dual-Graph Aggregation Mechanism**    *Global Graph Linking* and *Local Graph Linking* phase process are aggregated with *Dual-Graph Aggregation Mechanism* to obtain the unified structured representation of nodes in the query-graph. First, we aggregate the neighbor information with the local attention scores $\beta_{i,j,t}$, and then apply a gate function to extract essential features among the key node self and the neighbor information. The process is formulated as

$$\boldsymbol{h}_{i,j}^{k^{\text{neigh}}} = \sum_{t=1}^{T} \beta_{i,j,t} \boldsymbol{h}_t^k, \quad \boldsymbol{h}_{i,j}^{k^{\text{self}}} = \boldsymbol{h}_j^k, \quad (7)$$

$$\text{gate}_{i,j} = \theta \left( \boldsymbol{W}_{ng} \left[ \boldsymbol{h}_{i,j}^{k^{\text{self}}}; \boldsymbol{h}_{i,j}^{k^{\text{neigh}}} \right] \right), \quad (8)$$

$$\boldsymbol{h}_{i,j}^k = (1 - \text{gate}_{i,j}) * \boldsymbol{h}_{i,j}^{k^{\text{self}}} + \text{gate}_{i,j} * \boldsymbol{h}_{i,j}^{k^{\text{neigh}}}, \quad (9)$$

where $\boldsymbol{h}_{i,j}^{k^{\text{neigh}}}$ represents the neighbor context vector and $\boldsymbol{h}_{i,j}^k$ indicates the $j$-th key node neighbor-aware feature toward $i$-th query node. Finally, each query node aggregates the structure-aware information from all key nodes with the global attention score $\alpha_{i,j}$:

$$\boldsymbol{h}_i^{q^{\text{new}}} = \sum_{j=1}^{n} \alpha_{i,j} \left( \boldsymbol{h}_{i,j}^k + \boldsymbol{R}_{ij}^E \right), \quad (10)$$

$$\text{gate}_i = \theta \left( \boldsymbol{W}_{\text{gate}} \left[ \boldsymbol{h}_i^q; \boldsymbol{h}_i^{q^{\text{new}}} \right] \right), \quad (11)$$

$$\boldsymbol{h}_i^{q^{\text{Aggr}}} = (1 - \text{gate}_i) * \boldsymbol{h}_i^q + \text{gate}_i * \boldsymbol{h}_i^{q^{\text{new}}}, \quad (12)$$

where $\text{gate}_i$ indicates how much information the query node should receive from the key-graph. Consequently, we obtain the final query node representation $\boldsymbol{h}_i^{q^{\text{Aggr}}}$ with the structure-aware information of the key-graph.

## 4    Experiments

In this section, we conduct experiments on the Spider dataset [34], the benchmark of cross-domain Text-to-SQL, to evaluate the effectiveness of our model.

### 4.1    Experiment Setup

**Dataset and Metrics**    The Spider has been so far the most challenging benchmark on cross-domain Text-to-SQL, which contains 9 traditional specific-domain datasets, such as ATIS [8], GeoQuery [36], WikiSQL [1], IMDB [30] etc. It is split into the train set (8659 examples), development set (1034 examples) and test set (2147 examples), which are respectively distributed across 146, 20 and 40 databases. Since the fair competition, the Spider official has not released the test set for evaluation. Instead, participants must submit the model to obtain the test accuracy for the official non-released test set through the submission scripts provided officially by Yu et al. [34].[3]

**Embedding Initialization**    The pre-trained methods initialize the input embedding of question words and tables/columns. Specifically, in terms of the pre-trained vector, GloVe [21] is a common choice for the embedding initialization. And regarding the pre-trained language model (PLM), BERT [10] is also the mainstream embedding initialization method. In detail, BERT-base, BERT-large are applied according to the model scale. Additionally, the specific-domain pre-trained language

---

[3]Only submit up to two models per submission (at least two months before the next submission).

Table 1: Accuracy results on the Spider development set and test set.

| Approach | Dev | Test | Approach | Dev | Test |
|---|---|---|---|---|---|
| GNN [3] | 40.7 | 39.4 | RATSQL-HPFT + BERT-large | 69.3 | 64.4 |
| Global-GNN [4] | 52.7 | 47.4 | YCSQL + BERT-large | - | 65.3 |
| IRNet v2 [11] | 55.4 | 48.5 | DuoRAT + BERT-large [23] | 69.4 | 65.4 |
| RATSQL [27] | 62.7 | 57.2 | RATSQL + BERT-large [27] | 69.7 | 65.6 |
| **SADGA** | **64.7** | - | **SADGA + BERT-large** | **71.6** | **66.7** |
| EditSQL + BERT-base [37] | 57.6 | 53.4 | ShadowGNN + RoBERTa [7] | 72.3 | 66.1 |
| GNN + Bertrand-DR [15] | 57.9 | 54.6 | RATSQL + STRUG [9] | 72.6 | 68.4 |
| IRNet v2 + BERT-base [11] | 63.9 | 55.0 | RATSQL + GraPPa [35] | **73.4** | 69.6 |
| RATSQL + BERT-base [27] | 65.8 | - | RATSQL + GAP [25] | 71.8 | 69.7 |
| **SADGA + BERT-base** | **69.0** | - | **SADGA + GAP** | 73.1 | **70.1** |

models, e.g., GAP [25], GraPPa [35], STRUG [9] are also applied for better taking advantage of prior Text-to-SQL knowledge. Due to the limited resources, we conducted experiments with four pre-trained methods, GloVe, BERT-base, BERT-large and GAP, to understand the significance of SADGA.

**Implementation**   We trained our models on one server with a single NVIDIA GTX 3090 GPU. We follow the original hyperparameters of RATSQL [27] that uses batch size 20, initial learning rate $7 \times 10^{-4}$, max steps 40,000 and the Adam optimizer [16]. For BERT, the initial learning rate is adjusted to $2 \times 10^{-4}$, and the max training step is increased to 90,000. We also apply a separate learning rate of $3 \times 10^{-6}$ to fine-tune BERT. For GAP, we follow the original settings in Shi et al. [25]. In addition, we stack **3**-layer SADGA followed by **4**-layer RAT. More details about the hyperparameters are included in Appendix D.

Table 2: The **BERT-large** accuracy results on Spider development set and test set compared to RATSQL by hardness levels defined by Yu et al. [34].

| Model | Easy | Medium | Hard | Extra Hard | All |
|---|---|---|---|---|---|
| *Dev:* | | | | | |
| RATSQL | 86.4 | **73.6** | 62.1 | 42.9 | 69.7 |
| **SADGA** | **90.3** | 72.4 | **63.8** | **49.4** | **71.6** |
| *Test:* | | | | | |
| RATSQL | 83.0 | 71.3 | **58.3** | 38.4 | 65.6 |
| **SADGA** | **85.1** | **72.1** | 57.0 | **41.7** | **66.7** |

### 4.2   Overall Performance

The exact match accuracy results are presented in Table 1. Almost all results of the baselines are obtained from the official leaderboard. Except, RATSQL [27] does not provide BERT-base as PLM results on the development set, we have experimented with official implementation. As shown as the table, the proposed SADGA model is competitive with the baselines in the identical sub-table. Specifically, regarding the development set, our raw SADGA, SADGA + BERT-base, SADGA + BERT-large and SADGA + GAP all outperform their corresponding baselines. And with the GAP enhancement, our model is competitive with RATSQL + GraPPa as well. While regarding the test set, our models, only available for the BERT-large one and the GAP one, also surpass their competitors. At the time of writing, our best model SADGA + GAP achieved 3rd on the overall leaderboard. Note that our focus lies in developing an efficient base model but not a specific solution for the Spider dataset.

To better demonstrate the effectiveness, our SADGA is evaluated on the development set and test set compared with RATSQL according to the parsing difficulty level defined by Yu et al. [34]. In the Spider dataset, the samples are divided into four difficulty groups based on the number of components selections and conditions of the target SQL queries. As shown in Table 2, our SADGA outperforms the baseline on the **Extra-Hard** level by 6.5% and 3.3% on the development set and test set, respectively, which implies that our model can handle more complicated SQL parsing. This is most likely due to the fact that SADGA adopts a unified dual graph modeling method to consider both the global and local structure of the question and schema, which is more efficient for capturing the complex semantics of questions and building more exactly linkings in hard cases. The result also indicates that SADGA and RATSQL achieved the best of **Medium** and **Hard** on the test set,

respectively, but in the development set it is switched. It is an interesting finding that SADGA and RATSQL are adversarial and preferential in **Medium** and **Hard** levels data. After the statistics, we found that the distribution of data in the **Medium** and **Hard** levels changed from the development set to the test set (**Medium** 43.1% to 39.9%, **Hard** 16.8% to 21.5%), which is one of the reasons. And another reason we guess is that the target queries for these two types of data are relatively close to each other. Both **Medium** and **Hard** levels are mostly the join queries, but the **Extra-Hard** level is mostly nested queries.

## 4.3 Ablation Studies

Table 3: Accuracy of ablation studies on the Spider development set by hardness levels.

| Model | Easy | Medium | Hard | Extra Hard | All |
|---|---|---|---|---|---|
| **SADGA** | **82.3** | **67.3** | **54.0** | **42.8** | **64.7** |
| w/o Local Graph Linking | 83.5(+1.2) | 64.8(-2.5) | 53.4(-0.6) | 38.6(-4.2) | 63.2(-1.5) |
| w/o Structure-Aware Aggregation | 83.5(+1.2) | 62.1(-5.2) | 55.2(+1.2) | 42.2(-0.6) | 62.9(-1.8) |
| w/o GraphAggr($\mathcal{G}_S, \mathcal{G}_Q$) | 83.1(+0.8) | 64.1(-3.2) | 52.3(-1.7) | 40.4(-2.4) | 62.9(-1.8) |
| w/o GraphAggr($\mathcal{G}_Q, \mathcal{G}_S$) | 79.0(-3.3) | 63.7(-3.6) | 50.0(-4.0) | 41.6(-1.2) | 61.5(-3.2) |
| Q-S Linking via Dual-Graph Encoding | 82.3(-0) | 63.7(-3.6) | 51.1(-2.9) | 45.2(+2.4) | 63.1(-1.6) |
| w/o Relation Node (replace with edge types) | 79.4(-2.9) | 63.5(-3.8) | 54.6(+0.6) | 40.4(-2.4) | 62.1(-2.6) |
| w/o Global Pooling (Eq. 3 and Eq. 4) | 82.7(+0.4) | 64.3(-3.0) | 54.0(-0) | 41.6(-1.2) | 63.5(-1.2) |
| w/o Aggregation Gate (Eq. 8, $\text{gate}_{i,j} = 0.5$) | 81.9(-0.4) | 60.1(-7.2) | 54.6(+0.6) | 40.4(-2.4) | 61.2(-3.5) |
| w/o Relation Feature in Aggregation ($\boldsymbol{R}_{ij}^E$) | 79.4(-2.9) | 64.3(-3.0) | 54.6(+0.6) | 41.6(-1.2) | 62.7(-2.0) |
| **SADGA + BERT-base** | **85.9** | **71.7** | **58.0** | **47.6** | **69.0** |
| w/o Local Graph Linking | 85.5(-0.4) | 69.5(-2.2) | 54.0(-4.0) | 42.8(-4.8) | 66.4(-2.6) |
| w/o Structure-Aware Aggregation | 85.9(-0) | 68.8(-2.9) | 57.5(-0.5) | 41.0(-6.6) | 66.5(-2.5) |

To validate the effectiveness of each component of SADGA, ablation studies are conducted on different parsing difficulty levels. The major model variants are as follows:

**w/o Local Graph Linking** Discard the *Local Graph Linking* phase (i.e., Eq. 6 ~ 9), which means $\boldsymbol{h}_{i,j}^k$ in Eq. 10 is replaced by $\boldsymbol{h}_j^k$. There is no structure-aware ability during the dual graph aggregation.

**w/o Structure-Aware Aggregation** Remove the whole structure-aware aggregation module to examine the effectiveness of our designed graph aggregation method.

Other fine-grain ablation experiments are also conducted on our raw SADGA. The ablation experimental results are presented in Table 3. As the table shows, all components are necessary to SADGA. According to the results on the **All** level, our models, no matter the raw SADGA or the one with BERT-base enhancing, decrease by about 1.5% and 1.8% or 2.6% and 2.5% while discarding the *Local Graph Linking* phase and the entire structure-aware aggregation method, which indicates the positive contribution to SADGA. Especially on the **Extra-Hard** level, discarding the *Local Graph Linking* and the aggregation respectively both lead to a large decrease of accuracy, suggesting that these two major components strongly help SADGA deal with more complex cases. Interestingly, on the **Easy** level, the results indicate that these two components have no or slight negative influence on our raw model. This phenomenon is perhaps due to the fact that the **Easy** level samples do not require capturing the local structure of our dual graph while building the question-schema linking, but the structure-aware ability is highly necessary for the complicated SQL on the **Extra-Hard** level. Regarding other fine-grain ablation experiments, we give a further introduction and discussion on these ablation variants in Appendix E.

## 4.4 Case Study

To further understand our method, in this section, we conduct a detailed analysis of the case in which the question is "*What is the first name of every student who has a dog but does not have a cat?*".

**Global Graph Linking Analysis** We show the alignment figure between question words and tables/columns on the *Global Graph Linking* phase when the question-graph acts as the query-graph. As shown in Figure 5, we can obtain the interpretable result. For example, the question word "student"

has a strong activation with the tables/columns related to the student, which helps better build the cross graph linking between the question and schema. Furthermore, we can observe that the column "pet_type" is successfully inferred by the word "dog" or "cat".

**Local Graph Linking Analysis** On the *Local Graph Linking* phase, we compute the attention between the query node and the neighbors of the key node, which allows question words (tables/columns) to attend to the specific structure of the schema-graph (question-graph). In Figure 6, two examples about the neighbor attention on the *Local Graph Linking* phase are presented. As shown in the upper part of the Figure 6, the column "first_name" of table "Student" attends to neighbors of word "name" in the question, where word "first" and word "student" obtain a high attention score, indicating that the column "first_name" attends to the specific structure inside the dashed box.

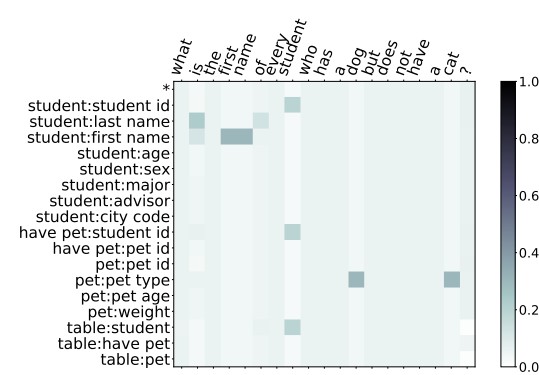

Figure 5: Alignment between question words and tables/columns on the *Global Graph Linking* phase.

Some tables/columns are difficult to be identified via matching-based alignment since they do not attend explicitly in the question, but they have a strong association with the question, e.g., table "Have_pet" in this case, which is also not identified in *Global Graph Linking*. Interestingly, as shown on the lower part of Figure 6 shows, table "Have_pet" acquires a high attention weight when the question word "student" attends to table "Student" and its neighbors. With the help of SADGA, the latent association between table "Have_pet" and word "student" can be detected, which corresponds exactly to the semantics of the question.

We also provide more samples in different database schemas compared to the baseline RATSQL and some corresponding discussions in Appendix F.

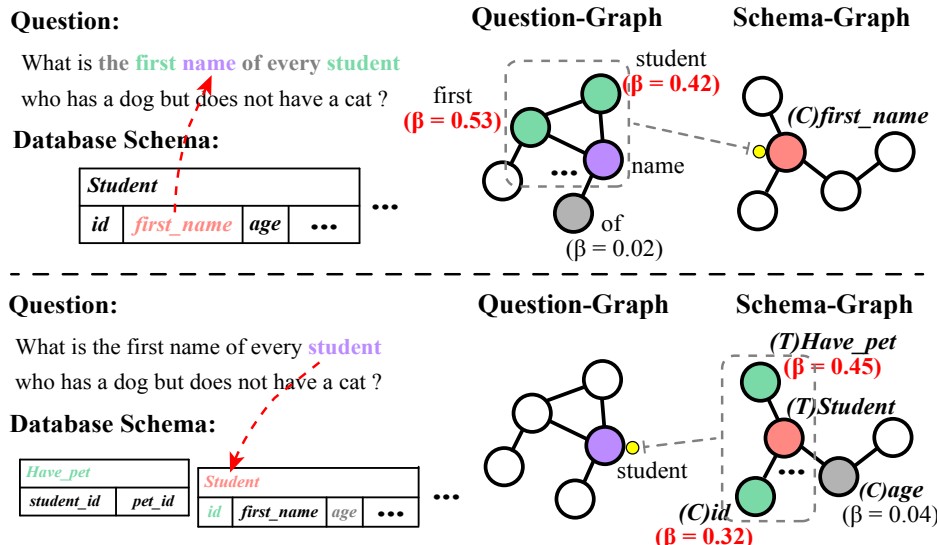

Figure 6: Analysis on the *Local Graph Linking* phase.

## 5 Related Work

**Cross-Domain Text-to-SQL** Recent architectures proposed for cross-domain Text-to-SQL show increasing complexity in both the encoder and the decoder. IRNet [11] encodes the question and

schema separately via LSTM with the string-match strategy and proposes to decode an abstracted intermediate representation (IR). RATSQL [27] proposes a unified encoding mechanism to improve the joint representation of question and schema. BRIDGE [19] serializes the question and schema into a tagged sequence and maximally utilizes BERT [10] and the database content to capture the question-schema linking. SmBoP [22] presents the first semi-autoregressive bottom-up semantic parser for the decoding phase in Text-to-SQL.

Besides, the graph encoder has been widely applied in cross-domain Text-to-SQL. Bogin et al. [3] is the first to encode the database schema using graph neural networks (GNNs). Global-GNN [4] applies GNNs to softly select a subset of tables/columns for the output query. ShadowGNN [7] presents a graph project neural network to abstract the representation of the question and schema. LGESQL [6] utilizes the line graph to update the edge features in the heterogeneous graph for Text-to-SQL, which further considers both local and non-local, dynamic and static edge features. Differently, our SADGA not only adapts a unified dual graph framework for both the question and database schema, but also devises a structure-aware graph aggregation mechanism to sufficiently utilize the global and local structure information across the dual graph on the question-schema linking.

**Graph Aggregation**    The global-local graph aggregation module [38] is proposed to model interactions across graphs and aggregate heterogeneous graphs into a holistic graph representation in the video titling task. Nevertheless, this graph aggregation method is only at the node level, i.e., it does not consider the structure during aggregation, indicating that nodes in the graph are a series of unstructured entities. Instead of simply using the node-level aggregation, our SADGA considers the local structure information in the aggregation process, contributing a higher-order graph aggregation method with structure-awareness.

**Pre-trained Models**    Inspired by the success of pre-trained language models, some recent works have tried to apply pre-trained objectives for text-table data. TAPAS [12] and TaBERT [32] leverage the semi-structured table data to enhance the representation ability of language models. For Text-to-SQL, GraPPa [35] is pre-trained on the synthetic data generated by the synchronous context-free grammar, STRUG [9] leverages a set of novel prediction tasks using a parallel text-table corpus to help solve the question-schema linking challenge. GAP [25] explores the direction of utilizing the generators to generate pre-trained data for enhancing the joint question and structured schema encoding ability. Moreover, Scholak et al. [24] proposes a method PICARD for constraining autoregressive decoders of pre-trained language models through incremental parsing.

## 6    Conclusions

In this paper, we propose a Structure-Aware Dual Graph Aggregation Network (SADGA) for cross-domain Text-to-SQL. SADGA not only introduces a unified dual graph encoding for both natural language question and database schema, but also devises a structure-aware aggregation mechanism of SADGA to take full advantage of the global and local structure information of the dual graph in the question-schema linking. Experimental results show that our proposal achieves 3rd on the challenging Text-to-SQL benchmark Spider at the time of writing. This study shows that both the dual-graph encoding and structure-aware dual graph aggregation method are able to improve the generalization ability of the cross-domain Text-to-SQL task. As future work, we will extend SADGA to other heterogeneous graph tasks and other alignment tasks.

## Acknowledgements

We thank Tao Yu and Yusen Zhang for their evaluation of our work in the Spider Challenge. We also thank the anonymous reviewers for their helpful comments. This research was supported in part by National Key R&D Program of China (2021ZD0111501), National Science Fund for Excellent Young Scholars (62122022), Natural Science Foundation of China (61876043, 61976052), Science and Technology Planning Project of Guangzhou (201902010058).

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
