# Supplementary Material:
# SADGA: Structure-Aware Dual Graph Aggregation Network for Text-to-SQL

**Ruichu Cai**[1,2]**, Jinjie Yuan**[1]**, Boyan Xu**[1]*****, Zhifeng Hao**[1,3]

[1] School of Computer Science, Guangdong University of Technology, Guangzhou, China
[2] Pazhou lab, Guangzhou, China
[3] College of Science, Shantou University, Shantou, China
cairuichu@gdut.edu.cn, yuanjinjie0320@gmail.com
hpakyim@gmail.com, zfhao@gdut.edu.cn

## A    Preliminaries

Gated Graph Neural Networks [4] and Relation-Aware Transformer [8] are two critical components of our proposed model. The preliminaries of these two components are introduced as follows.

### A.1   Gated Graph Neural Network

Gated Graph Neural Networks (GGNNs) have been proposed by Li et al. [4], which adopt the Gated Recurrent Unit (GRU) [2] layer to encode the nodes in graph neural networks. Given a graph $G = (V, E, T)$ including nodes $v_i \in V$ and directed label edges $(v_s, t, v_d) \in E$ where $v_s$ denotes the source node, $v_d$ denotes the destination node, and $t \in T$ denotes the edge type. The process of GGNN computing the representation $\boldsymbol{h}_i^{(l)}$ at step $l$ for the $i$-th node on $G$ is divided into two stages. First, aggregating the neighbor node representation $\boldsymbol{h}_k^{(l-1)}$ of $i$-th node, formulated as

$$\boldsymbol{f}_i^{(l)} = \sum_{t \in T} \sum_{(i,k) \in E_t} (\boldsymbol{W}_t \boldsymbol{h}_k^{(l-1)} + \boldsymbol{b}_t), \tag{1}$$

where $W_t$ and $b_t$ are trainable parameters for each edge type $t$. Second, aggregated vector $\boldsymbol{f}_i^{(l)}$ will be fed into a vanilla GRU layer to update the node representation at last step $\boldsymbol{h}_i^{(l-1)}$, noted as:

$$\boldsymbol{h}_i^{(l)} = \text{GRU}\left(\boldsymbol{h}_i^{(l-1)}, \boldsymbol{f}_i^{(l)}\right). \tag{2}$$

### A.2   Relation-Aware Transformer

Relation-Aware Transformer (RAT) [8] is an extension of Transformer [7], which introduces prior relation knowledge to the self-attention mechanism. Given a set of inputs $X = \{\boldsymbol{x}_i\}_{i=1}^n$ where $\boldsymbol{x}_i \in R^d$ and relation representation $\boldsymbol{r}_{ij}$ between any two elements $\boldsymbol{x}_i$ and $\boldsymbol{x}_j$ in $X$. The RAT layer (consisting of $H$ heads attention) can output an updated representation $\boldsymbol{y}_i$ with relational information for $\boldsymbol{x}_i$, formulated as

$$e_{i,j}^{(h)} = \frac{\boldsymbol{x}_i \boldsymbol{W}_Q^{(h)} \left(\boldsymbol{x}_j \boldsymbol{W}_K^{(h)} + \boldsymbol{r}_{ij}^K\right)^T}{\sqrt{d_z/H}}, \alpha_{i,j}^{(h)} = \text{softmax}_j \left\{ e_{i,j}^{(h)} \right\}, \tag{3}$$

$$\boldsymbol{z}_i^{(h)} = \sum_{j=1}^n \alpha_{i,j}^{(h)} \left(\boldsymbol{x}_j \boldsymbol{W}_V^{(h)} + \boldsymbol{r}_{ij}^V\right), \boldsymbol{z}_i = \text{Concat}(\boldsymbol{z}_i^{(1)}, ..., \boldsymbol{z}_i^{(H)}), \tag{4}$$

$$\tilde{\boldsymbol{y}}_i = \text{LayerNorm}(\boldsymbol{x}_i + \boldsymbol{z}_i), \boldsymbol{y}_i = \text{LayerNorm}(\tilde{\boldsymbol{y}}_i + \text{FC}(\text{ReLU}(\text{FC}(\tilde{\boldsymbol{y}}_i)))), \tag{5}$$

---

*Corresponding author

35th Conference on Neural Information Processing Systems (NeurIPS 2021).

where $h$ is head index, $\boldsymbol{W}_Q^{(h)}, \boldsymbol{W}_K^{(h)}, \boldsymbol{W}_V^{(h)} \in R^{d \times (d/H)}$ are trainable parameters, FC is a fully-connected layer, and LayerNorm is layer normalization [1]. Here $\alpha_{i,j}^{(h)}$ means that the attention score between $\boldsymbol{x}_i$ and $\boldsymbol{x}_j$ of head $h$.

## B  Relations of Dual-Graph Construction

All predefined relations used in the construction of the dual-graph and the cross-graph relations are summarized in Table 1.

Table 1: The predefined relations for Dual-Graph Construction.

|  | Node A | Node B | Predefined Relation |
|---|---|---|---|
| **Question-Graph Construction** | Word | Word | 1-order Word Distance
2-order Word Distance
Parsing-based Dependency |
| **Schema-Graph Construction** | Column | Column | Same Table Match
Primary-Foreign Key |
|  | Column | Table | Foreign Key
Primary Key
Table-Column Match |
|  | Table | Table | Primary-Foreign Key |
| **Cross-Graph** | Word | Table | Exact String Match
Partial String Match |
|  | Word | Column | Exact String Match
Partial String Match
Value Match |

The predefined relations of Question-Graph are summarized as follows:

- **1-order Word Distance** Word A and word B are adjacent to each other in the question.
- **2-order Word Distance** Word A and word B are spaced one word apart in the question.
- **Parsing-based Dependency** The specific grammatical relation between word A and word B generated by the Stanford CoreNLP toolkit [5].

The predefined relations of Schema-Graph are summarized as follows:

- **Same Table Match** Both column A and column B belong to the same table.
- **Primary-Foreign Key (Column-Column)** Column A is a foreign key for a primary key column B of another table.
- **Foreign Key** Column A is a foreign key of table B.
- **Primary Key** Column A is a primary key of table B.
- **Table-Column Match** Column A belongs to table B.
- **Primary-Foreign Key (Table-Table)** Table A has a foreign key column for a primary key column of table B.

The predefined relations of Cross-Graph are summarized as follows:

- **Exact String Match (Word-Table)** Word A is part of table B, and the question contains the name of table B.
- **Partial String Match (Word-Table)** Word A is part of table B, and the question does not contain the name of table B.
- **Exact String Match (Word-Column)** Word A is part of column B, and the question contains the name of column B.
- **Partial String Match (Word-Column)** Word A is part of column B, and the question does not contain the name of column B.
- **Value Match** Word A is part of the cell values of column B.

## C    Decoder Details

The decoder in our model aims to output a sequence of rules (actions) that generates the corresponding SQL syntax abstract tree (AST) [9]. Given the final representations $\boldsymbol{h}^q$, $\boldsymbol{h}^t$ and $\boldsymbol{h}^c$, of the question words, tables and columns respectively from the encoder. Let $\boldsymbol{h} = [\boldsymbol{h}^q; \boldsymbol{h}^t; \boldsymbol{h}^c]$. Formally,

$$\Pr(P \mid \boldsymbol{h}) = \prod_t \Pr\left(\text{Rule}_t \mid \text{Rule}_{<t}, \boldsymbol{h}\right), \tag{6}$$

where $\text{Rule}_{<t}$ are all the previous rules. We apply an LSTM [3] to generate the rule sequence. The LSTM hidden state $\boldsymbol{H}_t$ and the cell state $\boldsymbol{C}_t$ at step $t$ are updated as:

$$\boldsymbol{H}_t, \boldsymbol{C}_t = \text{LSTM}\left(\boldsymbol{I}_t, \boldsymbol{H}_{t-1}, \boldsymbol{C}_{t-1}\right). \tag{7}$$

Similar to Wang et al. [8], the LSTM input $\boldsymbol{I}_t$ is constructed by:

$$\boldsymbol{I}_t = [\boldsymbol{r}_{t-1}; \boldsymbol{z}_t; \boldsymbol{e}_t; \boldsymbol{r}_{pt}; \boldsymbol{H}_{pt}], \tag{8}$$

where $\boldsymbol{r}_{t-1}$ is the representation of the previous rule, $\boldsymbol{z}_t$ is the context vector calculated using the attention on $\boldsymbol{H}_{t-1}$ over $\boldsymbol{h}$, and $\boldsymbol{e}_t$ is the learned representation of the current node type. In addition, $pt$ is the step corresponding to generating the parent node in the AST of the current node.

With the LSTM output $\boldsymbol{H}_t$, all rule scores at step $t$ are calculated. The candidate rules are either schema-independent, e.g., the grammar rule, or schema-specific, e.g., the table/column. For the schema-independent rule $u$, we compute its score as:

$$\Pr(\text{Rule}_t = u \mid \text{Rule}_{<t}, \boldsymbol{h}) = \text{softmax}_u\left(L(\boldsymbol{H}_t)\right), \tag{9}$$

where $L$ is a 2-layer MLP with the *tanh* activation. To select the table/column rule, we first build the alignment matrices $\boldsymbol{M}^T$, $\boldsymbol{M}^C$ between entities (question word, table, column) and tables, columns respectively with the relation-aware attention as a pointer mechanism:

$$\overline{\boldsymbol{M}}_{i,j}^T = \boldsymbol{h}_i \boldsymbol{W}_Q^t (\boldsymbol{h}_j^t \boldsymbol{W}_K^t + \boldsymbol{R}_{ij}^E)^T, \boldsymbol{M}_{i,j}^T = \text{softmax}_j\left\{\overline{\boldsymbol{M}}_{i,j}^T\right\}, \tag{10}$$

$$\overline{\boldsymbol{M}}_{i,j}^C = \boldsymbol{h}_i \boldsymbol{W}_Q^c (\boldsymbol{h}_j^c \boldsymbol{W}_K^c + \boldsymbol{R}_{ij}^E)^T, \boldsymbol{M}_{i,j}^C = \text{softmax}_j\left\{\overline{\boldsymbol{M}}_{i,j}^C\right\}, \tag{11}$$

where $\boldsymbol{M}^T \in R^{(|q|+|t|+|c|)\times|t|}$, $\boldsymbol{M}^C \in R^{(|q|+|t|+|c|)\times|c|}$. Then, we calculate the score of the $j$-th column/table:

$$\overline{\alpha}_i = \boldsymbol{H}_t \boldsymbol{W}_Q \left(\boldsymbol{h}_i \boldsymbol{W}_K\right)^T, \alpha_i = \text{softmax}_i\left\{\overline{\alpha}_i\right\}, \tag{12}$$

$$\Pr(\text{Rule}_t = \text{Table}[j] \mid \text{Rule}_{<t}, \boldsymbol{h}) = \sum_{i=1}^{|q|+|t|+|c|} \alpha_i \boldsymbol{M}_{i,j}^T, \tag{13}$$

$$\Pr(\text{Rule}_t = \text{Column}[j] \mid \text{Rule}_{<t}, \boldsymbol{h}) = \sum_{i=1}^{|q|+|t|+|c|} \alpha_i \boldsymbol{M}_{i,j}^C. \tag{14}$$

## D    Hyperparameters

The hyperparameters of our model under different pre-trained models are listed in Table 2.

## E    Fine-grained Ablation Studies

Due to page limitations, we cannot further discuss the fine-grained ablation studies in the main paper. Therefore, the fine-grained ablation studies are discussed in this section. Firstly, all the ablation variants are presented in detail as follows:

**w/o Local Graph Linking**    Discard the *Local Graph Linking* phase (Eq. 6 ~9), i.e., $\boldsymbol{h}_{i,j}^k$ in Eq. 10 is replaced by $\boldsymbol{h}_j^k$. There is no structure-aware ability during the dual graph aggregation.

**w/o Structure-Aware Aggregation**    Remove the entire Structure-Aware Aggregation module in SADGA to examine the effectiveness of our designed graph aggregation method.

Table 2: Hyperparameters for GloVe, BERT-base, BERT-large and GAP setting.

| Hyper-paramter | GloVe | BERT-base | BERT-large | GAP |
|---|---|---|---|---|
| Size | 300 | 768 | 1024 | 1024 |
| Batch size | 20 | 24 | 24 | 24 |
| Max step | 40k | 90k | 81k | 61k |
| Learning rate | 7.44e-4 | 3.44e-4 | 2.44e-4 | 1e-4 |
| Learning rate scheduler | Warmup polynomial | Warmup polynomial | Warmup polynomial | Warmup polynomial |
| Warmup steps | 2k | 10k | 10k | 5k |
| Bert learning rate | - | 3e-6 | 3e-6 | 1e-5 |
| Clip gradient | - | 2 | 1 | 1 |
| Number of SADGA layers | 3 | 3 | 3 | 3 |
| Number of RAT layers | 4 | 4 | 4 | 4 |
| RAT heads | 8 | 8 | 8 | 8 |
| Number of GGNN layers | 2 | 2 | 2 | 2 |
| SADGA dropout | 0.5 | 0.5 | 0.5 | 0.5 |
| RAT dropout | 0.1 | 0.1 | 0.1 | 0.1 |
| Encoder hidden dim | 256 | 768 | 1024 | 1024 |
| Decoder LSTM size | 512 | 512 | 512 | 512 |
| Decoder dropout | 0.21 | 0.21 | 0.21 | 0.21 |

**w/o GraphAggr$(\mathcal{G}_S, \mathcal{G}_Q)$**  Remove the aggregation process from the question-graph $\mathcal{G}_Q$ to the schema-graph $\mathcal{G}_S$ in Structure-Aware Aggregation, signifying that the nodes in the schema-graph could not obtain the structure-aware information from the question-graph.

**w/o GraphAggr$(\mathcal{G}_Q, \mathcal{G}_S)$**  Similar to w/o GraphAggr$(\mathcal{G}_S, \mathcal{G}_Q)$.

**Q-S Linking via Dual-Graph Encoding**  In contrast to variant **w/o Structure-Aware Aggregation**, which removes the entire aggregation module in SADGA, we preserve the predefined cross-graph relations during dual-graph encoding. This variant guarantees the ability of question-schema (Q-S) linking, and its performance variation better reflects the contribution of Structure-Aware Aggregation.

**w/o Relation Node (replace with edge types)**  Remove the relation node in Dual-Graph Encoding. Regrading how to use the information of the prior relationship in the question-graph and schema-graph, we represent the predefined relations with the edge types, introducing more trainable parameters.

**w/o Global Pooling (Eq. 3 and Eq. 4)**  Remove the global pooling step during the Structure-Aware Aggregation, i.e., Eq. 3 and Eq. 4, to examine whether the global information of the query-graph is helpful for graph aggregation.

**w/o Aggregation Gate (Eq. 8)**  Discard the gate mechanism between the global information and the local information in *Dual-Graph Aggregation Mechanism*. Instead of the gating mechanism, we average the weight of the global information and the local information, i.e., $\text{gate}_{i,j} = 0.5$ in Eq. 8.

**w/o Relation Feature in Aggregation $(R_{ij}^E)$**  Remove the cross-graph relation bias between the question word and table/column in the attention step of Structure-Aware Aggregation. This model variant does not utilize any predefined cross-graph relations.

As shown in Table 3 (Table 3 of the main paper), all the components are necessary to SADGA. Regarding **w/o Local Graph Linking** and **w/o Structure-Aware Aggregation**, we have discussed these two major ablation variants in detail in the main paper. When compared to **w/o Structure-Aware Aggregation**, SADGA gets worse results when it retains one-way aggregation, i.e., **w/o GraphAggr$(\mathcal{G}_S, \mathcal{G}_Q)$** and **w/o GraphAggr$(\mathcal{G}_Q, \mathcal{G}_S)$**. We guess that this observation occurs because the update of dual graph node representation is imbalanced in one-way aggregation. The downgraded performance of **Q-S Linking via Dual-Graph Encoding** better demonstrates the necessity and effectiveness of our proposed structure-aware aggregation method for question-schema linking. The downgraded performance of **w/o Relation Node** is due to the increase of relational

Table 3: Accuracy of ablation studies on Spider development set by hardness levels.

| Model | Easy | Medium | Hard | Extra Hard | All |
|---|---|---|---|---|---|
| **SADGA** | **82.3** | **67.3** | **54.0** | **42.8** | **64.7** |
| w/o Local Graph Linking | 83.5(+1.2) | 64.8(-2.5) | 53.4(-0.6) | 38.6(-4.2) | 63.2(-1.5) |
| w/o Structure-Aware Aggregation | 83.5(+1.2) | 62.1(-5.2) | 55.2(+1.2) | 42.2(-0.6) | 62.9(-1.8) |
| w/o GraphAggr($\mathcal{G}_S, \mathcal{G}_Q$) | 83.1(+0.8) | 64.1(-3.2) | 52.3(-1.7) | 40.4(-2.4) | 62.9(-1.8) |
| w/o GraphAggr($\mathcal{G}_Q, \mathcal{G}_S$) | 79.0(-3.3) | 63.7(-3.6) | 50.0(-4.0) | 41.6(-1.2) | 61.5(-3.2) |
| Q-S Linking via Dual-Graph Encoding | 82.3(-0) | 63.7(-3.6) | 51.1(-2.9) | 45.2(+2.4) | 63.1(-1.6) |
| w/o Relation Node (replace with edge types) | 79.4(-2.9) | 63.5(-3.8) | 54.6(+0.6) | 40.4(-2.4) | 62.1(-2.6) |
| w/o Global Pooling (Eq. 3 and Eq. 4) | 82.7(+0.4) | 64.3(-3.0) | 54.0(-0) | 41.6(-1.2) | 63.5(-1.2) |
| w/o Aggregation Gate (Eq. 8, $\text{gate}_{i,j} = 0.5$) | 81.9(-0.4) | 60.1(-7.2) | 54.6(+0.6) | 40.4(-2.4) | 61.2(-3.5) |
| w/o Relation Feature in Aggregation ($\boldsymbol{R}_{ij}^E$) | 79.4(-2.9) | 64.3(-3.0) | 54.6(+0.6) | 41.6(-1.2) | 62.7(-2.0) |
| **SADGA + BERT-base** | **85.9** | **71.7** | **58.0** | **47.6** | **69.0** |
| w/o Local Graph Linking | 85.5(-0.4) | 69.5(-2.2) | 54.0(-4.0) | 42.8(-4.8) | 66.4(-2.6) |
| w/o Structure-Aware Aggregation | 85.9(-0) | 68.8(-2.9) | 57.5(-0.5) | 41.0(-6.6) | 66.5(-2.5) |

edge type, which leads to the increase of trainable parameters. The downgraded performance of **w/o Aggregation Gate** indicates the advantages of the gated-based aggregation mechanism, which provides the flexibility to filter out useless local structure information. The downgraded performance of **w/o Global Pooling** indicates that the global information of question-graph or schema-graph is beneficial to another graph. Our SADGA **w/o Relation Feature in Aggregation** is comparable with RATSQL [8] (62.7%), which reflects the effectiveness of the structure-aware aggregation method to learn the relationship between the question and database schema without relying on prior relational knowledge at all.

## F   Case Study Against Baseline

In Figure 1, We show some cases generated by our SADGA and RATSQL [8] from the **Hard** or **Extra Hard** level samples of Spider Dataset [10]. Both SADGA and RATSQL are trained under the pre-trained model GAP [6]. In Case 1 and Case 2, RATSQL misaligned the word "museum" and "rank", resulting in the incorrect selection of tables and columns in the generated query. RATSQL utilizes the predefined relationship based on a string matching strategy to cause the above misalignment problem. Our SADGA is able to link the question words and tables/columns correctly in the hard cases of multiple entities, which is beneficial from the local structural information introduced by the proposed structure-aware aggregation method. In Cases 3∼6, RATSQL generates semantically wrong query statements, especially when the target is a complex query, such as a nested query. Compared with RATSQL, SADGA adopts a unified dual-graph modeling method to consider both the global and local structure of the question and schema, which is more efficient for capturing the complex semantics of questions and building more exactly linkings in hard cases.

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

**(1)**   **Question:**   What are the id, name and membership level of visitors who have spent the largest amount of money in total in all museum tickets?

**Gold SQL:**   SELECT T2.visitor_id , T1.name, T1.level_of_membership FROM Visitor AS T1 JOIN Visit AS T2 ON T1.id = T2.visitor_id GROUP BY T2.visitor_id ORDER BY Sum(T2.total_spent) DESC LIMIT 1.

**RATSQL Result:**   SELECT Museum.museum_id, Museum.name, Visitor.level_of_membership FROM Museum JOIN Visit JOIN Visitor GROUP BY Museum.museum_id ORDER BY Sum(Visit.total_spent) Desc LIMIT 1. ✗

**SADGA Result:**   SELECT Visitor.id, Visitor.name, Visitor.level_of_membership FROM Visit JOIN Visitor ON Visit.visitor_id = Visitor.id GROUP BY Visitor.id ORDER BY Sum(Visit.total_spent) Desc LIMIT 1. ✓

**(2)**   **Question:**   Find the first name, country code and birth date of the winner who has the highest rank points in all matches.

**Gold SQL:**   SELECT T1.first_name , T1.country_code , T1.birth_date FROM Players AS T1 JOIN matches AS T2 ON T1.player_id = T2.winner_id ORDER BY T2.winner_rank_points DESC LIMIT 1

**RATSQL Result:**   SELECT Players.first_name, Players.country_code, Players.birth_date FROM Players JOIN Rankings ON Players.player_id = Rankings.player_id ORDER BY Rankings.ranking_points Desc LIMIT 1. ✗

**SADGA Result:**   SELECT Players.first_name, Players.country_code, Players.birth_date FROM Players JOIN Matches ON Players.player_id = Matches.winner_id ORDER BY Matches.winner_rank_points Desc LIMIT 1. ✓

**(3)**   **Question:**   Find all airlines that have flights from both airports 'APG' and 'CVO'.

**Gold SQL:**   SELECT T1.airline FROM Airlines AS T1 JOIN Flights AS T2 ON T1.id = T2.airline WHERE T2.source_airport = "APG" INTERSECT SELECT T1.airline FROM Airlines AS T1 JOIN Flights AS T2 ON T1.id = T2.airline WHERE T2.source_airport = "CVO".

**RATSQL Result:**   SELECT Airlines.airline FROM Flights WHERE Flights.source_airport = 'VALUE' INTERSECT SELECT Airlines.airline FROM Flights WHERE Flights.source_airport = 'VALUE' . ✗

**SADGA Result:**   SELECT Airlines.airline FROM Airlines JOIN Flights ON Airlines.id = Flights.airline WHERE Flights.source_airport = 'VALUE' INTERSECT  SELECT Airlines.airline FROM Airlines JOIN Flights ON Airlines.id = Flights.airline WHERE Flights.source_airport = 'VALUE' . ✓

**(4)**   **Question:**   What are the names of all stadiums that did not have a concert in 2014 ?

**Gold SQL:**   SELECT name FROM Stadium EXCEPT SELECT T2.name FROM Concert AS T1 JOIN Stadium AS T2 ON T1.stadium_id = T2.stadium_id WHERE T1.year = 2014.

**RATSQL Result:**   SELECT Stadium.name FROM Stadium WHERE Stadium.stadium_id NOT IN (SELECT Concert.stadium_id FROM Concert WHERE Concert.year = 'VALUE' ). ✗

**SADGA Result:**   SELECT Stadium.name FROM Stadium EXCEPT SELECT Stadium.name FROM Stadium JOIN Concert ON Stadium.stadium_id = Concert.stadium_id WHERE Concert.year = 'VALUE' . ✓

**(5)**   **Question:**   Show name of all students who have some friends and also are liked by someone else.

**Gold SQL:**   SELECT T2.name FROM Friend AS T1 JOIN Highschooler AS T2 ON T1.student_id = T2.id INTERSECT SELECT T2.name FROM Likes AS T1 JOIN Highschooler AS T2 ON T1.liked_id = T2.id.

**RATSQL Result:**   SELECT Highschooler.name FROM Highschooler WHERE Friend.friend_id IN (SELECT Likes.student_id FROM Likes). ✗

**SADGA Result:**   SELECT Highschooler.name FROM Highschooler JOIN Friend ON Friend.student_id = Highschool.id INTERSECT SELECT Highschooler.name FROM Highschooler JOIN Likes ON Highschooler.id = Likes.liked_id. ✓

**(6)**   **Question:**   What is the name of the semester with no students enrolled?

**Gold SQL:**   SELECT semester_name FROM Semesters WHERE semester_id NOT IN (SELECT semester_id FROM Student_Enrolment).

**RATSQL Result:**   SELECT Semesters.semester_name FROM Semesters EXCEPT SELECT Semesters.semester_name FROM Semesters JOIN Student_Enrolment ON Semesters.semester_id = Student_Enrolment.semester_id. ✗

**SADGA Result:**   SELECT Semesters.semester_name FROM Semesters WHERE Semesters.semester_id NOT IN (SELECT Student_Enrolment.semester_id FROM Student_Enrolment). ✓

Figure 1: More cases at the **Hard** or **Extra Hard** level in different database schemas. (RATSQL + GAP vs. SADGA + GAP)