# OpenReview forum: "SADGA: Structure-Aware Dual Graph Aggregation Network for Text-to-SQL"
_NeurIPS.cc/2021/Conference — NeurIPS 2021 Poster_

### Official Review · Reviewer_1Eab · 2021-07-15

**Rating:** 7
**Confidence:** 4

**Summary:**

In this paper, the authors propose Structure-Aware Dual Graph Aggregation Network (SADGA), a system designed for cross-domain Text-to-SQL task.

As input, the SADGA system takes a question (in language) and a database schema. A question graph and a schema graph are subsequently extracted by following a set of rules (14 pre-defined structural relations). Then, a Gated Graph Neural Network (GGNN)-based dual encoder encodes both graphs by performing message propagation, resulting their corresponding graph representations that are aware of the defined relations and the graph structures.

Next, an aggregation module is applied. In specific, it consists of three sub-modules:

1. An attention-based global graph linking layer: given graph A and B, this layer learns the mapping relationship between each node in A with all nodes in B. This layer captures global structure information

2. An attention-based local graph linking layer: given a node $N_i$ in graph A and a node $N_j$ in graph B, the layer is trained to capture the relationship between $N_i$ with all $N_j$'s neighbors. This can be seen as capturing the local structure information.

3. A gate mechanism-based aggregator: it learns to what extent, a node should be represented by its own encoding, or by the weighted sum of the other graph, in which, the weights are computed by either the global or local graph linking layer.

After the aggregation module, the representations are flattened and further guides the generation by a decoder.

On the standard benchmark for cross-domain Text-to-SQL, Spider, the proposed model achieve decent performance. A set of ablation study further show the effectiveness of the proposed SADGA system.


**Limitations And Societal Impact:**

I do not see obvious negative societal impact given the current version of this work.

**Main Review:**


## Overall comments

The task is interesting and can be very useful product wise.

The proposed methods make sense. The approach of converting both input modalities into graph, and then performing all representation computations (encoding, attention, aggregation etc.) in the graph space seems novel and hopeful.

The authors provide a set of well-designed experiments (including main results, ablation study, and qualitative plus quantitative analyses) with convincing results.

The paper is clearly written and easy to follow.


## Concerns and questions

**(Q1.)** It would be nice if the authors can emphasize how is SADGA particularly designed for  **cross-domain** Text-to-SQL task. I understand that Spider is the standard benchmark for assessing such systems, but in addition to showing a better validation and test scores, does the model really improves by somewhat alleviate the difficulty of cross-domain. It would be better if such example and analyses are provided.

**(Q2.)** As a reader without too much experience on the Spider dataset, I think it would be very helpful to describe a bit more about the dataset. For instance, what do the difficulty levels mean? Furthermore with that, the author can explain, for example, why in Table 3 that SADGA outperforms baseline more significantly on the two most difficult sets.

**(Q3.)** The overall system seems to get rid of the question text completely. It that possible the question (in its language form) can remain helpful in some of the model steps? I'm asking because as shown in Table 1, the word-word relations do not seem to capture all information from the question (e.g., word ordering and maybe some other syntactical structures).


## Typos and minor things:

1. In the main paper, what happens after the aggregation needs clarifying. Specifically, Section 3 ends at $\mathcal{G}'_Q$ and $\mathcal{G}'_S$ (Eqn. 1), what happens next can only be found from Figure 2. For readers not familiar with the baseline pipeline, this can be a bit difficult to understand.

2. Typo in L204: SADGGA

3. Typo in L207: Util

**Time Spent Reviewing:**

5

---

> ### Author Response · Authors · 2021-08-10
> **Discussion about the contribution and experiments**
>
> Thanks for your positive comments and valuable suggestions. The detailed response is as follows.
>
> **Concerns and questions:**
>
> **(Q1.)**  *It would be nice if the authors can emphasize how is SADGA particularly designed for **cross-domain** Text-to-SQL task. I understand that Spider is the standard benchmark for assessing such systems, but in addition to showing a better validation and test scores, does the model really improves by somewhat alleviate the difficulty of cross-domain. It would be better if such example and analyses are provided.*
>
> **Answer:** Thanks for your suggestion.
>
> Regarding "how is SADGA particularly designed for cross-domain Text-to-SQL task": SADGA is designed to address the main challenge of cross-domain Text-to-SQL task, i.e., improving the generalization ability of the trained question-schema linking model to the unseen databases. Focusing on this challenge, first we adapt to a unified graph neural network encoder to model both the NL question and database schema to reduce the structure gap from NL side; second, we further devise a structure-aware aggregation method based on the unified graph modeling to learn the linking between the question-graph and schema-graph on local structure level, instead of simply linking question and schema on node-level based on predefined relationships.
>
> Regarding "It would be better if such examples and analyses are provided": We have already provided some examples in Section E of the submitted supplementary materials. As shown in Case 1 and Case 2, RATSQL falsely links the word "museum" to the table Museum (Case 1) and the word "rank" to the table Rankings (Case 2) in the generated query; while SADGA correctly builds the question-schema linking through the proposed structure-aware aggregation method. We hope this case study is able to show our effect on addressing the generalization problem in the cross-domain task. We also recognize that the examples may not be enough to show the contribution of our proposed model, and we will try to provide more examples and discussions in the revised manuscript.
>
> **(Q2.)**  *As a reader without too much experience on the Spider dataset, I think it would be very helpful to describe a bit more about the dataset. For instance, what do the difficulty levels mean? Furthermore with that, the author can explain, for example, why in Table 3 that SADGA outperforms baseline more significantly on the two most difficult sets.*
>
> **Answer:** Thanks for the suggestion. We will try to provide more details of the dataset in the revised manuscript.
>
> Regarding "what do the difficulty levels mean?", the difficulty levels are determined by the Spider dataset. In the Spider dataset, the samples are divided into four difficulty groups based on the number of components selections and conditions of the target SQL queries.
>
> Regarding "SADGA outperforms baseline more significantly on the two most difficult sets", this is because SADGA adopts a unified dual-graph modeling method to consider both the structure of the question and the structure of the database schema, which is more efficient to capture the semantics of natural language questions in hard cases.
>
> **(Q3.)**  *The overall system seems to get rid of the question text completely. It that possible the question (in its language form) can remain helpful in some of the model steps? I'm asking because as shown in Table 1, the word-word relations do not seem to capture all information from the question (e.g., word ordering and maybe some other syntactical structures).*
>
> **Answer:** Thank you for your insightful suggestion. Yes, we agree that complete word sequence information may be effective for representation learning. As an improvement, we will try to use LSTM to pre-encode the sequential ordering information of the problem to get a better initialization of SADGA. We will provide the above options in the released source code.
>
> **Typos and minor things:**
>
> **Answer:** We will polish the representation to solve all these issues. Thanks for your careful review!

---

> > ### Comment · Reviewer_1Eab · 2021-08-25
> > **Thanks**
> >
> > Thanks for providing the response. The rebuttal has answered most of my concerns and questions. I will keep my score of 7.

---

### Official Review · Reviewer_RQfk · 2021-07-16

**Rating:** 7
**Confidence:** 4

**Summary:**

This paper introduces SADGA, a method for cross-domain text-to-SQL. First, a question-graph and schema-graph is constructed. Second, a unified dual-graph neural network is proposed to dually encode the question and database schema into respective graph encodings. Importantly, a structure-aware dual graph aggregation method is proposed for question-schema linking, wherein global graph linking maps the relationship between query nodes and the key graph's global structure using relation-aware attention, local graph linking maps the attention of query nodes to neighbours of key-graph nodes, and dual-graph aggregation unifies the attention scores into a final node representation. Experiments and ablation studies are conducted on the Spider benchmark.

**Limitations And Societal Impact:**

Yes.


**Main Review:**

**Originality**

This paper is well situated in previous work and explores interesting directions suggested by the methods of existing literature. Unifying the question and schema has been explored with string matching (IRNet), and predefined relations (RAT). Graph encoding of questions or schemas have been separately utilised. The unique contribution of this paper is in aggregating graph encodings to unify question and schema representations.

Global-local graph aggregation is not widely utilised, and the authors leverage this method in a novel way for text-to-SQL while improving the aggregation mechanism.

**Quality**

The methods proposed are appropriately justified and are rational proposals for addressing question-schema linking.The work is complete with well formulated specifications of the proposed encoding and aggregation methods.

The comparisons with other models used in their experiments on the Spider dataset are enlightening as they highlight the convincing improvements of their method in different aspects (unification of question-schema, enhancement with BERT, and augmentation with pre-trained domain-specific models). Moreover, the breakdown of results into complexity of text-to-SQL parsing required provides a justified picture into how their method improves on complicated SQL parsing.

The ablation study of the paper is particularly strong, and justifies the necessity of their proposed local graph linking and aggregation mechanism. Moreover, the authors are upfront about how the added structure-aware ability reduces performance on easier parsing examples.

- Do the authors think that tuning the saliency of local structure during question-schema linking is a possible method to address the tradeoffs in performance when parsing easier SQL queries?

- Why does SADGA + BERT not exhibit this tradeoff on easier SQL queries?

**Clarity**

The paper is written and structured well. The provided case study in Section 4.4 is particularly useful for helping readers to understand.

**Significance**

The results show a marked improvement on the Spider dataset, suggesting that dual-graph aggregation is a good method for addressing question-schema linking. The semantic parsing community will clearly benefit this result.

**Time Spent Reviewing:**

4

---

> ### Author Response · Authors · 2021-08-10
> **Discussion about tuning the saliency of the local structure**
>
> Thanks for your positive comments and valuable suggestions. The detailed response is as follows.
>
> **(Q1.)**  *Do the authors think that tuning the saliency of local structure during question-schema linking is a possible method to address the tradeoffs in performance when parsing easier SQL queries?*
>
> **Answer:** Thanks for the insightful suggestion. We agree that tuning the saliency of local structures is a possible solution for this phenomenon. This is also our motivation to use gate functions to aggregate the local structure, which tries to tune the importance of the local structure adaptively. We also agree that the problem still exists after using the above gate mechanism, as the model is trained on the whole training set containing all difficulty levels. We will consider this problem as future work.
>
> **(Q2.)**  *Why does SADGA + BERT not exhibit this tradeoff on easier SQL queries?*
>
> **Answer:** Thanks for your careful review. A possible reason is that the better pre-trained language model, BERT, brings more stability to SADGA. We will try to investigate this problem in future work.

---

> > ### Comment · Reviewer_RQfk · 2021-09-02
> > **Thank you**
> >
> > Thank you for the response. My score remains at 7.

---

### Official Review · Reviewer_Db2V · 2021-07-16

**Rating:** 6
**Confidence:** 3

**Summary:**

This work proposes a new model, SADGA, for the cross-database Text-to-SQL task. The model represent the input natural language question as a graph (question-graph) and the input database schema as a graph (schema-graph). The model then encodes and aggregates these two graphs jointly (structure-aware dual graph aggregation). The model achieves substantially improved results on the Spider benchmark.

**Limitations And Societal Impact:**

While I do not see limitations and societal impact discussed in the paper, I think that the proposed technique for text-to-SQL is not likely to have negative societal impact.

**Main Review:**

Reasons to Accept:
- Cross-database generalization is an important research problem. The proposed idea of unifying and better aligning the representations of question and database schema is quite interesting.
- Experimental results are strong: the ablation study shows the effect of the proposed SADGA module, and the final system achieves 3rd place in the Spider benchmark.

Issues to resolve
- It would be great to improve the clarify of the introduction. In Figure 1, does the "strong"/"weak" attention mean they are actual attention results computed in trained models? (If so, what exact models does "the existing methods" refer to?)  It would also be helpful to mention what "latent association" (e.g. L41) refers to in Figure 1.
- It is a bit hard to see how controlled the experiments are. It appears that RAT-SQL is the closest baseline architecture, and the model proposed in this paper adds the SADGA module as shown in Figure 2. It would be great to discuss how many additional parameters the SADGA module introduces compared to RAT-SQL. If it uses substantially more parameters, it is worthwhile to do a more controlled experiment, e.g. running a baseline model that has the architecture of RAT-SQL but uses the same amount of parameters as the proposed model, or a baseline model that uses basic MLPs in place of the SADGA module.
- Related work discussion is a bit light. For instance, the main technique of this work is to use a graph to unify two modalities, text and database schema, and there are several related works of this idea - unifying text and knowledge base via graph (e.g. https://arxiv.org/abs/1809.00782, https://arxiv.org/abs/1909.05311), unifying text and code via graph (e.g. https://arxiv.org/abs/2005.10636). It may be nice to discuss the conceptual commonality and difference with these techniques.

Typos/grammar:
- L22: "achieved a great progress"?
- L34: "global" -> "globally"?
- L40: "such a structural gap hinders the trained model from being efficiently adapted to..."
- L44: "problem" -> "question" (use "question" consistently?)
- L49: "adapts to" -> "adopts"?
- L51: "cross" -> "across"
- Figure3 caption: "attends not only" "attends to not only"

**Time Spent Reviewing:**

3

---

> ### Author Response · Authors · 2021-08-10
> **Response and discussions about the introduction and experiments**
>
> Thanks for your positive comments and valuable suggestions. The detailed response is as follows.
>
> **Issues to resolve:**
>
> - *It would be great to improve the clarify of the introduction. In Figure 1, does the "strong"/"weak" attention mean they are actual attention results computed in trained models? (If so, what exact models does "the existing methods" refer to?) It would also be helpful to mention what "latent association" (e.g. L41) refers to in Figure 1.*
>
>   **Answer:** Thanks for the suggestion. We will polish the presentation to improve the clarity in the revised manuscript.
>
>   Regarding "does the "strong"/"weak" attention means they are actual attention results computed in trained models?": Yes, the "strong"/"weak" attention means the attention results computed in trained models.
>
>   Regarding "what exact models do "the existing methods" refer to?": As mentioned in L31-35, the existing methods refer to the matching-based methods, e.g., IRNet, or the learning-based methods, e.g., RAT-SQL.
>
>   Regarding "It would also be helpful to mention what "latent association" (e.g., L41) refers to in Figure 1": We are sorry that there is no "latent association" in the figure due to our carelessness. We will solve this problem in the revised manuscript. Specifically, latent associations are the relations that cannot be extracted by string matching or word meaning. For example, as shown in Figure 5, the table Have\_pet is highly associated with the question word "student​" but cannot be extracted by string matching.
>
> - *It is a bit hard to see how controlled the experiments are. It appears that RAT-SQL is the closest baseline architecture, and the model proposed in this paper adds the SADGA module as shown in Figure 2. It would be great to discuss how many additional parameters the SADGA module introduces compared to RAT-SQL. If it uses substantially more parameters, it is worthwhile to do a more controlled experiment, e.g. running a baseline model that has the architecture of RAT-SQL but uses the same amount of parameters as the proposed model, or a baseline model that uses basic MLPs in place of the SADGA module.*
>
>   **Answer:** Thanks for the discussion. We will provide more details from the following three aspects:  1) for the number of parameters, RATSQL has 13M parameters, and SADGA contains 15M parameters, i.e., the number of parameters are comparable; 2) for the model architecture, RAT-SQL have 8 RAT layers and SADGA have 3 SADGA layers and 4 RAT layers as mentioned in the paper (L196-197), i.e., the complexity of the architecture is comparable; 3) for the pre-defined relations, RAT-SQL has more than 50 predefined relations, while SADGA only needs about 10 relations, i.e., our model is easier to use. In summary, our model has comparable complexity as RAT-SQL but is easier to use.
>
> **About the related works, typos/grammar and the quality of the presentation:**
>
> ​	**Answer:** We will polish the presentation to solve these issues. Thanks for your careful review and your provided related works!

---

> > ### Comment · Reviewer_Db2V · 2021-08-25
> > **Thanks to authors**
> >
> > Thank you for the detailed response to my questions/suggestions - they make sense to me.  I will keep the rating of 6.

---

### Official Review · Reviewer_hu28 · 2021-07-19

**Rating:** 6
**Confidence:** 4

**Summary:**

The paper deals with the Text-to-SQL problem. Different to previous approaches, the paper frames the problem as the encoding of two graphs: one graph representing the natural-language query and one graph representing the database schema. They process the graph independently first and then in conjunction considering the local and global structure of both graphs and use this information to generate the output SQL query. Combined with powerful pre-trained embeddings, their proposal got the third place in a challenging public benchmark.

**Limitations And Societal Impact:**

YES

**Main Review:**

The problem tackled is certainly relevant and the experimental results provide clear evidence of the good practical properties of the proposal: the best version of the model got the 3rd place (now 4th) in the Spider benchmark.

The authors did a good job showing the empirical impact of their design decisions showing that each one of them helps to obtain the good results reported, although it would be great to see a bit more (see Question 1 below).

One of the strongest points from my point of view is that their core proposal (SAGDA) without any external pre-training methods added, outperforms other core proposals (RAT-SQL, IRNet). Actually the difference in the leader boards appeared to be driven by the additions to every proposal specially on the usage of pretraining models for embedding computation (either using BERT, GAP, GraPPa, etc.).

On the negative side, I think that the novelty of the proposal is somehow limited. I like the idea of representing the input query and the schema as graphs and treating them as such, but as explained by the authors in the related work, this is not entirely novel. The input encoding (unsing BERT or GAP) as well as the final decoding (using RAT and an LSTM decoding) are somehow standard and borrowed from previous work. Node embeddings in graphs are computed with GNNs. Thus what is left is the local and global combination of information from the two graphs plus the aggregation of this information. For this combination and aggregation the authors combine standard gadgets (pooling, gating, attention, and so on). Thus from an architecture point of view, there is not much novelty.

Besides novelty, I think that the biggest weakness of the paper is the lack of clarity in the description of the proposal. I think that the proposal is simple enough to be easily described, but the authors didn't manage to effectively do that. Even being familiar with the topic, it was not easy to grasp the main points and separate what the authors are proposing from what is previous work. The figures doesn't help either (some specific comments below). Sometimes I was more confused than enlightened by the figures presented. Moreover, I think that it would be difficult even to an expert in the field, to reproduce the proposal by just reading the paper.

To summarize, I think that the contribution is somehow marginal but valuable. At least from a practical the proposal is effective. Nevertheless the presentation is not effective enough and that is why I think that the paper is not ready yet for publication.

Some specific comments:

- In Figure 1, it is difficult to understand "the message" of the figure. What does it try to say? maybe having two parts of the graph in the upper right, one for student and another for professor, would be better (with different attentions in each case). Another part that make difficult to follow the figures is the use of colors to differentiate weak and strong attention. At a first glance to the figure, both seems equal and I was wondering why in the right part it is ok to have the arrow that was not ok to have in the left part.

- When firts reading the paper, it was not totally clear to me if the graph construction from the query and the schema was deterministic or part of some learning. Although it is clear later by the more detailed description, it would be good to explicitly stating that at the beginning of the description.

- line 73-74, what is "the natural" structure of the database schema? I think that it would be good to explicitly describe what the authors use as this structure.

- line 74-75, the authors mentioned a "Gated GNN" to initially encode the question and schema graphs, but I think that that GNN is not explained in the paper. You should warn the reader if you are using some standard part here.

- lines 78-82, I think that the word "relation" is misleading here. At the beginning I was reading it as in "relational database" where relation == table, but relation means another thing here so you should be careful explaining that.

- lines 83-87, I really don't understand here if you use a tree or a sequence to represent the output. It seems that you use a sequential representation of a tree structure, but the wording is a bit confusing. Please be precise and clear on the final part of your model.

- line 107, what is "1-order word distance"? do you mean that words that are at distance 1 are connected in the graph? please be clear here (similar with "2-order word distance").

- lines 121-125 (Dual-Graph Encoding). This is an important part of the proposal and it is very difficult to understand what is happening here. I can only infer that you have two graphs, say G1=(V1, E1) and G2=(V2,E2) and that you construct a third graph D = (V1 U V2, E1 U E2 U E') where E' is a set of new edges of the form (v1,v2) with v1 in V1 and v2 in V2. If that is the case, please consider explaining it in this terms. Some mathematical notation to clarify will be great here. Also please clearly explain what E' is (again I infer that this is somehow constructed from what you describe in Table 1, but I'm not sure).

- Equations in Section 3.3. I think that the equations are OK and one can understand what is happening here, but it would be great to factorize them inside some general functions, as you did with GraphAggr in equations (1) and (2). That is, to abstract what are the input and outputs to some modules, put a clear name for the module, add equations combining the modules, and after that give the details of each module. Given that you have 11 equations in total (3-13), it would be great to guide the reader by factorizing groups of equations into abstract functions. For instance when updating h_j^k in equations 3-5, you can name it as h_j^k := GatedUpdate(MeanPool(G_q), h_j^k). Notice that you are doing not exactly the same but similar later in equations 8-10 and in equations 11-13.

- Regarding the use of pre-training embeddings, I think that you should be more precise on how you actually put them to work with your architecture. This should be an easy addition (a couple of sentences would suffice) and would help someone trying to reproduce your method. Also please be precise and clear on whether you fine-tune BERT and other pretrained methods as part of your architecture.


Specific questions:

Q1: About Table 4, do you know what is the impact of global and local when SAGDA when combined with BERT-large? with GAP? I think that it would be a nice addition to have those rows at the end of Table 4.

=== AFTER REBUTTAL

I have increased my score (5 --> 6).

**Time Spent Reviewing:**

6

---

> ### Author Response · Authors · 2021-08-10
> **The main contribution is the structure-awareness aggregation, but not the encoding, decoding...**
>
> Thanks for your valuable comments and suggestions. The detailed response is as follows.
>
> **The negative side:**
>
> *Novelty: I think that the novelty of the proposal is somehow limited. I like the idea of representing the input query and the schema as graphs and treating them as such, but as explained by the authors in the related work, this is not entirely novel. The input encoding (unsing BERT or GAP) as well as the final decoding (using RAT and an LSTM decoding) are somehow standard and borrowed from previous work. Node embeddings in graphs are computed with GNNs. Thus what is left is the local and global combination of information from the two graphs plus the aggregation of this information. For this combination and aggregation the authors combine standard gadgets (pooling, gating, attention, and so on). Thus from an architecture point of view, there is not much novelty.*
>
> **Answer:** We are sorry that we do not agree with your comments on the novelty. Generally, our main contribution is the structure-aware dual graph aggregation method, which explores the local structure to build the linking between the database schema and the question, which is totally different from the existing node level alignment/matching based methods. Specifically, we will clarify this point from the following point-to-point responses.
>
> Regarding "the idea of representing the input query and the schema as graphs and treating them as such", some existing Text-to-SQL works applied the graph encoder for DB schema but sequential encoder for the question. Instead of using the multi-structure encoder, we propose a unified dual-graph neural network to encode the question-graph and schema-graph to reduce the structure gap in the encoding process.
>
> Regarding "this combination and aggregation the authors combine standard gadgets (pooling, gating, attention, and so on)", we want to clarify our contribution lies in the proposed structure-aware dual graph aggregation to take full advantage of the global and local structure information of the dual graph for better linking the question and schema. The key idea of the proposed method is not only based on the global node level alignment or matching, but the combination of the local structure level aggregation process.
>
> Regarding some methods you mention that is somehow standard and borrowed from previous work, we want to clarify that they are not the contributions of this work. (i) Pretrained language models (BERT or GAP) have been widely used in embedding-initialization to enhance the performance of Text-to-SQL and other NLP tasks. (ii) The tree-structured decoder of Yin and Neubig [26] is already a general practice (RAT-SQL [23], Global-GNN [3], ShadowGNN [5]).
>
>
>
> **Some specific comments:**
>
> - *In Figure 1, it is difficult to understand "the message" of the figure. What does it try to say? maybe having two parts of the graph in the upper right, one for student and another for professor, would be better (with different attentions in each case). Another part that make difficult to follow the figures is the use of colors to differentiate weak and strong attention. At a first glance to the figure, both seems equal and I was wondering why in the right part it is ok to have the arrow that was not ok to have in the left part.*
>
>   **Answer:** The message of Figure 1 is as follows.
>
>   The left part is about the existing approaches, which usually treat the question as a sequence and apply string matching or attention mechanism (based on the prior matching relation) to build the question-schema linking. In this case, the question word "name" is likely to have a strong linking/attention with both the column First\_name of the table Student and the column First_name of the table Professor (the red arrow, strong attention/linking), which seems reasonable but actually unsuitable. In fact, the word "name" should be only linked to the table Student.
>
>   The right part is about our method, which explores the local structure to help build the linking. First, to eliminate the structure gap between question and schema during linking, we treat both the question and schema as the graph structure. Second, we use structure-aware graph aggregation framework SADGA to build the linking between the two graphs. Hence, SADGA successfully removes the candidate linking between the word "name" and the column First\_name of table Professor (the green arrow, weak attention/linking).
>
> - *When firts reading the paper, it was not totally clear to me if the graph construction from the query and the schema was deterministic or part of some learning. Although it is clear later by the more detailed description, it would be good to explicitly stating that at the beginning of the description.*
>
>   **Answer:** The graph construction of question-graph and schema-graph is deterministic. We have provided the details of the graph construction in Section 3.1. We will try to clarify this point in the revised version.
>
> - *line 73-74, what is "the natural" structure of the database schema? I think that it would be good to explicitly describe what the authors use as this structure.*
>
>   **Answer:** "The natural" structure of the database refers to the database-specific relationship such as primary key match, foreign key match, table-column match, which are listed in Schema-Graph Construction of Table 1. We are sorry for the vague word "natural", and we would like to explicitly describe what we use as this "natural" structure here in the revised manuscript.
>
> - *lines 83-87, I really don't understand here if you use a tree or a sequence to represent the output. It seems that you use a sequential representation of a tree structure, but the wording is a bit confusing. Please be precise and clear on the final part of your model.*
>
>   **Answer:** The details are as follows: first, the decoder generates an abstract syntax tree; then the abstract syntax tree is transformed to the sequential SQL query. We also want to emphasize that this workflow is widely used in the existing text-to-SQL methods.
>
> - *line 107, what is "1-order word distance"? do you mean that words that are at distance 1 are connected in the graph? please be clear here (similar with "2-order word distance").*
>
>   **Answer:** Yes, "1-order word distance" means that words that are at distance 1 are connected in the graph, i.e., in the natural language question, two words that are adjacent to each other have this relation (the “2-order word distance” is similar).  We will revise the manuscript to clarify these points.
>
> - *lines 121-125 (Dual-Graph Encoding). This is an important part of the proposal and it is very difficult to understand what is happening here. I can only infer that you have two graphs, say G1=(V1, E1) and G2=(V2,E2) and that you construct a third graph D = (V1 U V2, E1 U E2 U E') where E' is a set of new edges of the form (v1,v2) with v1 in V1 and v2 in V2. If that is the case, please consider explaining it in this terms. Some mathematical notation to clarify will be great here. Also please clearly explain what E' is (again I infer that this is somehow constructed from what you describe in Table 1, but I'm not sure).*
>
>   **Answer:** We are sorry that there are some misunderstandings. Specifically, 1) In L122-123, “performing message propagation among their self-structure before building the two-stage graph linking cross dual-graph” means that we encode the Question-Graph and Schema-Graph **separately** before aggregation. 2) We introduce relation nodes to link the nodes. Concretely, if node A and node B have a relation R, then we introduce an extra node R into the graph and link R to both A and B using undirected edges. 3) There is no edge across the two graphs. 4) In addition, each relation node is initialized to a learnable vector. We will provide more details about the encoding in the revised manuscript.
>
> - *Equations in Section 3.3. I think that the equations are OK and one can understand what is happening here, but it would be great to factorize them inside some general functions, as you did with GraphAggr in equations (1) and (2) ... Notice that you are doing not exactly the same but similar later in equations 8-10 and in equations 11-13.*
>
>   **Answer:** Thank you very much for the excellent suggestion. We will revise the manuscript as you suggested.
>
> - *Regarding the use of pre-training embeddings, I think that you should be more precise on how you actually put them to work with your architecture. This should be an easy addition (a couple of sentences would suffice) and would help someone trying to reproduce your method. Also please be precise and clear on whether you fine-tune BERT and other pretrained methods as part of your architecture.*
>
>   **Answer:** Regarding "how to put the pre-training embeddings to our architecture", the pre-training embeddings are directly used as the initialization of the nodes before the Dual-Graph Encoding stage, similar to most of the works in the Text-to-SQL task. We have fine-tuned the pre-trained language model and provided the detailed fine-tune hyperparameters in the supplementary material. And the source code will be released if our work is accepted.
>
> **Specific questions:**
>
> **Q1:** *About Table 4, do you know what is the impact of global and local when SAGDA when combined with BERT-large? with GAP? I think that it would be a nice addition to have those rows at the end of Table 4.*
>
> **Answer:** Thanks for the suggestion. We are sorry that we have not investigated the impact of global and local in the ablation study with BERT-large or GAP due to the limited resources. In fact, we only choose the lighter pre-trained embeddings (GloVe and BERT-base) in the ablation study. We will try to conduct more experiments as you suggested in the revised manuscript.
>
> **About the quality and clarity of the presentation:**
>
> We will polish the presentation to solve these issues. Thanks for your careful review!

---

> > ### Comment · Reviewer_hu28 · 2021-08-15
> > **Thank you for you response.**
> >
> > Thank you for you response. I will increase my score accordingly.

---

### Author Response · Authors · 2021-09-03
**Thanks**

Dear Reviewers,

We are grateful for your careful reading and consideration of our response! Thanks for your patience.

Best regards,
Authors of Submission 7668

---

### Decision · Program_Chairs · 2021-09-28

**Decision:**

Accept (Poster)

**Comment:**

This paper proposes SADGA, a method for improving cross-domain Text-to-SQL, where the model has to generalize to unseen database schemas. The core method proposes to parse both natural language query and the database schema and then compute an alignment between both graphs which is then passed to the module producing SQL output.  The assumption is that learning the alignment between query graph and database schema is more transferrable/robust across novel database schemas. The paper shows improvement on cross-domain TextToSQL benchmark (Spider).

--

Reviewers are positive about this paper. Although query and schema graphs have already been used in the literature for Text-to-SQL, the central idea of aligning query and schema graphs via local-global aggregation is considered novel and promising. Overall, the model achieves strong empirical performance. One reviewer noted how SAGDA outperforms other models without any pre-trained component, which strengthens the claim of the paper. The main criticisms revolve around clarity of exposition and lack of interpretation. Authors promised in the rebuttal to add interpretable examples of how SAGDA solves the cross-domain TextToSQL benchmark. I additionally suggest the authors to strengthen the main motivation (e.g. matching in graph-space leads to more transferrable model's behaviour). Figure 1 needs some work (e.g. longer and more explanatory caption) as well as the intro needs proof-reading. Authors responded to reviewers' concerns about clarity issues and I believe all the concerns can be easily addressed in the revised version.

Overall, this paper provides a well-executed empirical contribution to cross-domain TextToSQL problem with positive empirical results. The graph-matching architecture proposed here can possibly inform and be extended to other tasks. Therefore, I recommend this paper for acceptance.

**Consistency Experiment:**

NeurIPS has a long history of experimentation. In 2014, NeurIPS ran an experiment in which 10% of submissions were reviewed by two independent committees to quantify the randomness in the review process. This year, we repeated a variant of this experiment to see how the quality of the review process has changed over time.  This paper was part of the experiment and was therefore assigned to two committees (consisting of reviewers, an Area Chair, and a Senior Area Chair) that reached independent decisions.  If both committees made the same recommendation, this recommendation was followed. If a single committee recommended acceptance, the paper was accepted (with the exception of a few cases in which the other committee identified what we considered a fatal flaw, e.g., an error in a key result).

This copy’s committee reached the following decision: **Accept (Poster)**

The other committee assigned to the paper recommended **Reject**.  You can find the other set of reviews, along with any follow up discussion with the authors here:
https://openreview.net/forum?id=NJg6R1ATGpe